# Identification of Heart Sounds with an Interpretable Evolving Fuzzy Neural Network

**DOI:** 10.3390/s20226477

**Published:** 2020-11-12

**Authors:** Paulo Vitor de Campos Souza, Edwin Lughofer

**Affiliations:** Department of Knowledge-Based Mathematical Systems, Johannes Kepler University Linz Altenberger Strasse 69, 4040 Linz, Austria; edwin.lughofer@jku.at

**Keywords:** evolving fuzzy neural network, heart murmur, SOF, pattern classification problem

## Abstract

Heart problems are responsible for the majority of deaths worldwide. The use of intelligent techniques to assist in the identification of existing patterns in these diseases can facilitate treatments and decision making in the field of medicine. This work aims to extract knowledge from a dataset based on heart noise behaviors in order to determine whether heart murmur predilection exists or not in the analyzed patients. A heart murmur can be pathological due to defects in the heart, so the use of an evolving hybrid technique can assist in detecting this comorbidity team, and at the same time, extract knowledge through fuzzy linguistic rules, facilitating the understanding of the nature of the evaluated data. Heart disease detection tests were performed to compare the proposed hybrid model’s performance with state of the art for the subject. The results obtained (90.75% accuracy) prove that in addition to great assertiveness in detecting heart murmurs, the evolving hybrid model could be concomitant with the extraction of knowledge from data submitted to an intelligent approach.

## 1. Introduction

Society seeks to work on solutions that reduce the negative impacts of groups of diseases on human beings routinely. Among these diseases, those that attack the heart stand out, and may decrease the standard of living when playing sports or daily activities, and even cause death [1]. Heart diseases are constant targets for applying artificial intelligence in solving problems arising from heart diseases, thereby identifying patterns that can be useful for medical teams to initiate treatments or surgical procedures [2].

A heart murmur is a comorbidity related to blood flow passage through the heart, generating sounds that can identify a circulatory problem regarding the heart’s functions. They can be identified in both children and adults, but for the most part, they can be just innocent murmurs with no other changes and with absolutely normal physical development. Another form of this heart murmur is due to complications of heart diseases that generally occurred in childhood, and care must be taken on the part of doctors so that the treatment to be carried out is appropriate for the maintenance of a healthy life on the patient’s part [3].

A heart murmur can exist in two ways: physiological or congenital; the second does not have a precise explanation for its cause. A heart murmur (congenital or acquired) needs special attention and adequate tests for its identification in time. Its main reasons for existence are linked to changes in the heart valves; communication between the aorta and the pulmonary artery; and rheumatic fever (caused by the body’s immune reaction against antigens or components of the streptococcus). It should be noted that degenerative diseases can also lead to a heart murmur [3].

The diagnosis of this disease is carried out by identifying heart sounds, comparing them with the frequency of familiar heart sounds. There are other forms, such as the electrocardiogram, chest X-rays and echocardiogram. However, this paper focuses on identifying heart murmurs by identifying sound patterns coming from the heart. As it is a disease that can be asymptomatic, the correct and effective diagnosis must be present in the routines of the patient and the medical team because in most severe cases of the disease, surgical interventions are fundamental for the maintenance of a good standard of life for the patient.

Several types of research are conducted jointly by artificial intelligence (AI) and the medical field. This type of partnership generates artifacts that can improve research or find fundamental patterns for improving the identification of diseases. Regarding the heart murmur problem, there is a public dataset with healthy heart sounds and a murmur. This dataset provided by Liu et al. [4] assists in constructing intelligent systems and techniques capable of assisting medical teams with diagnosing a heart murmur. Therefore, this dataset supported several surveys carried out to determine anomalies in heart sounds.

Intelligent approaches in detecting heart murmur are frequent in science, as in Rud et al. [5], which uses support vector machines to detect heart murmurs. Quiceno-Manrique et al. [6] and Mukherjee et al. [7] use a feature selection approach to define the best features to correctly classify heart murmurs using phonocardiographic signals or wavelet transform [8] proposed by Kumar et al. and Avendano-Valencia et al. [9], which uses time-frequency representations to define features about the problem.

Despite the growing amount of research in the area of identification of cardiac sounds, few models can bring assertive answers about the problem, and at the same time, extract knowledge about the evaluated data. With rare exceptions (such as the work of Soares et al. [10]), there are few logical relationships generated through the data submitted to the models. To solve this deficiency, hybrid models based on neural networks and fuzzy systems are used for two main processes in parallel: to obtain a high accuracy for the problem while extracting knowledge through fuzzy rules. This paper proposes the creation of an evolving three-layer model capable of solving problems related to heart murmurs. The first layer of the model builds Gaussian neurons, where a self-organizing technique [11] obtains the centers and sigmas based on empirical operators [12]. The first layer neurons’ weights are defined by a leave-one-feature-out (LOFO) technique, which is based on a separability criterion measuring the impact of each feature in terms of its contribution to the discriminatory power of the whole feature set [13]. In the second layer, in the definition of fuzzy rules, logical fuzzy neurons are used to aggregate neurons in the first layer. A neural aggregation network is interconnected with the second layer through output weights estimated by a weighted recursive least squares technique, with weights equal to the fuzzy rules’ activation degrees generated in the second layer of the model. The neural aggregation network is responsible for the process of defuzzification and for obtaining answers about the identification of heart murmurs.

This paper expects to use an evolving fuzzy neural network based on logical neurons (EFNN-LN) to identify cardiac problems based on sound evaluations of the heart. Unlike related works (as discussed above), our approach is able to extract linguistically readable fuzzy rules from the data and thus allows insight for the experts (new knowledge, better understanding of the process, etc.)—which we will demonstrate in the results section by showing the extracted rules for the two classes (heart murmur or no heart murmur) in if–then form. Thus, one main contribution of this paper is the extraction of readable rules to understand heart murmur problems (in which situations they occur, i.e., in which combinations of feature value ranges). Another contribution is the increased readability of rules by providing feature weights: features with low weights can be ignored and thus the rule length reduced → more compact representation; furthermore, the impacts of features in the rules may give clues for experts as to which features can be seen as more and which ones as less important to describe the classes. Furthermore, the fuzzy neurons are allowed to be constructed through advanced aggregation operators, apart from conventionally used t-norms in current related evolving fuzzy neural network approaches. This induces more flexibility in terms of allowing both AND * and OR connections of the rule antecedent parts (where in conventional approaches typically only AND connections are induced).

The paper is composed as follows: Section 2 presents the main theoretical concepts about the essential components in our model. The ideas of the evolving fuzzy neural network architecture, the fuzzy neuron and all aspects related to incremental, evolving training procedures of our neuro-fuzzy system’s components and parameters will be demonstrated. In Section 3, information about the dataset to be used in the experiment is presented to the reader. In Section 4, all the results in comparison with related SoA works and terms of achieved rules (interpretation) are presented to solve the heart sound problems. Finally, Section 5 highlights the conclusions about the study.

## 2. Evolving Fuzzy Neural Networks and Our Approach: EFNN-LN

Fuzzy neural networks [14] are hybrid models that work in solving problems with knowledge extraction. These intelligent models can extract knowledge through fuzzy rules based on the dataset evaluated by the model. The synergy between artificial neural networks that provide forms of training to solve problems, and fuzzy systems, which foster arguments to facilitate problems’ interpretability, works to solve complex problems of different natures, mainly in the health area. Hybrid models are present in several intelligent algorithms for solving complex problems. They work in synergy through the use of the most considerable advantages in the techniques used. Fuzzy neural network models and neuro-fuzzy networks are relevant highlights for this category [15].

These models work with several classes of problems related to robotics, time series prediction, linear regression, dynamic systems, etc. Their main difference compared to traditional intelligent models is the ability to extract knowledge from rules. This knowledge base extracted from the data can be verified with professionals experienced in solving these problems so that this set of rules can create expert systems or promote the dissemination of knowledge in companies through training on the aspects raised by the hybrid models [16]. The evolving fuzzy neural networks follow the same architectural logic as the conventional fuzzy neural networks; however, the parameters, fuzzification and defuzzification methods are based on adaptive and evolving algorithms for updating parameters and gaining new knowledge on demand and on the fly whenever new data samples are submitted to the training procedure [17]. This can be usually done in a single-pass manner [18], thereby allowing the models to be incrementally trained from (fast) streaming data [19]. This opens up enormous flexibility when system state or patients behaviors change (which typically result in concept drift issues [20]), as these changes can be quickly integrated without using old data (so no time-intensive re-training is required). These models have already been used to solve problems in the health area—for instance, take Iglesias et al. with the work on recognition of human activities during daily life [21,22], and Malcangi et al. who used an evolving hybrid model to solve problems in seismocardiograms to control and report heartbeats in everyday activities [23]. Another paper that uses evolving hybrid approaches is related to stroke control proposed by Kasabov et al. [24]. Other work related to performance in the health area can be seen in two recent surveys proposed by de Campos Souza [16] and Škrjanc et al. [17].

This paper’s evolving hybrid model has three layers, where the first is responsible for the fuzzification process. The second layer is responsible for building the fuzzy rules and training the model. It is also noteworthy that the first two layers of the model are seen as a fuzzy inference system, capable of transforming data into knowledge. Finally, the third layer is responsible for the defuzzification process through the weights of the fuzzy rules combined with the model inputs. This layer is made up of artificial neurons with linear activation functions and can be seen as a neural aggregation network. Figure 1 schematically presents the model architecture proposed in this paper. Next, the subsequent steps for constructing the fuzzy neural network with evolution based on logical neurons (EFNN-LN) will be defined.

### 2.1. First Layer—Fuzzyfication Process and Gaussian Neurons

The first layer is composed by *L* fuzzy neurons built by the fuzzification process described in the hereafter. All fuzzy sets created during the input space granulation are considered Gaussians, with the modal values measured as the cloud centers’ projections in the corresponding dimensions. This process of determining neurons in the first layer follows the premises defined in the models reported in [25]. However, a fixed number of neurons is used, which is pre-parameterized according to trial and error tuning phases. The use of our evolving approach opens up the possibility to autonomously seek the adequate number of neurons (according to the nature of the data stream to describe the classes sufficiently, etc.) to minimize the exponential relationship impact of fuzzy neurons created between the number of membership functions chosen for the model and the number of features in the dataset. As a result, it should create more compact models (neurons are evolved when actually needed).

The automatic definition of intelligent model architecture is not a new issue. It has been the subject of study mainly in the neural architecture search (NAS) [26], a technique proposed by Zoph and Le [27] and Baker et al. [28] that works on intelligent models’ architectural definitions. They use techniques such as search space (set of operations such as convolutions or fully connected networks) to form valid network architectures or define previous architectures through optimization techniques and finally evaluation strategies to verify the model’s efficiency in solving problems. The evolving fuzzy neural network used in this paper uses the same idea of the automatic definition of architecture based on data. However, it uses the fuzzification process defined by data density to build neurons representing interpretability to the problem. Thus, it can be noted that the difference between the NAS and the proposed model is that the architectural definition of the NAS is based on the optimal network to solve the problem, whereas in our evolving fuzzy neural network, the input data are transformed into fuzzy sets. These result in a model architecture which is linguistically interpretable, as can be see in the results section, where there are examples showing the rules learned from real-world heart sounds classification problems. The neurons in the model proposed in the paper, differently from the NAS approach, also define the best set of neurons to make the problem interpretable. Furthermore, our approach is an adaptive and evolving approach, meaning that it can elegantly act on data streams or additional datasets by updating the model solely with new samples, without requiring (possibly time-intensive) re-training phases from scratch. To the best of our knowledge, this has not been considered in NAS approaches so far.

Another difference from previous related works is related to the weights of Gaussian neurons. Previously published works defined the weights to construct the neurons at random in the interval between 0 and 1. However, this does not facilitate interpretability, because for different training runs different weights will be extracted, leading to non-deterministic results. This may be also the case when the weights are just randomly initialized, but further learned from the data, because different initializations often lead to different (local) solutions, as finding a global unique solution within an (numerical) optimization context is for many (real-world, noisy) datasets an unresolved issue. To solve this situation, a deterministic approach based on an advanced separability criterion will be used in this paper. It calculates feature weights from the data $w_{il}$ (for *i* = 1 ... *n* and *l* = 1 ... *L*), which depend on the importance of features according to their contributions to the discriminatory power of the whole feature set with respect to the classes contained in the data. The discriminatory power will measured by an extended statistical criterion based on Fisher’s seperability criterion, which is known to be deterministic.

For training the first layer, we evaluate the samples and number of dimensions of the dataset, where for every input variable $x_{i}$, *L* fuzzy sets are represented by $A_{i}^{l},l=1,\;\ldots ,\;L$, whose membership functions are the activation functions of the corresponding neurons. Thus, the outputs of the initial layer are the membership degrees correlated with the input values, i.e., $a_{il}=\nu \left(A_{i}^{l}\right)$ for i=1,…,n and l=1,…,L, where *n* is the number of inputs and *L* represents the number of fuzzy sets (neurons) for each input. The centers’ coordinate-wise values and the dispersions (σ) define these sets, which are obtained from the clouds formed by the evolving approach.

Therefore, it is expected that the Gaussian neurons of the first layer will be built according to the fuzzification process. In this paper, the Gaussian neurons of the first layer are expressed by:(1)$$a_{il}\left(x_{i},c_{il},\sigma_{il}\right)=e^{-\frac{1}{2}{\left(\frac{x_{i}-c_{il}}{\sigma_{il}}\right)}^{2}},\;for\;j=1\ldots \;n,\;l=1\;\ldots \;L$$
where $c_{il}$ and $\sigma_{il}$ are respectively the center and width of the *l*th Gaussian function for input variable *i*.

#### 2.1.1. Fuzzification Approach: Self-Organising Fuzzy Logic

Fuzzification approaches are extremely important for the operation of hybrid models, especially fuzzy neural networks. This step is responsible for transforming discrete inputs from datasets into fuzzy outputs, with different degrees of relevance to fuzzy sets representing linguistic terms. From this process, the factors to compose the Gaussian neurons projected in the centers and sigmas are obtained through the fuzzification procedure.

In this paper, a non-parametric approach to fuzzification proposed in Gu and Angelov [11] is used. The self-organizing technique identifies clouds from the observed data through an offline training process, allowing the fuzzy neural network model’s architecture to represent the assessed problem’s data. Unlike the model’s initial approach, in this paper, we will only use the stage of identification of clouds through empirical data analytics operators (EDA), discarding the procedures related to the classifier based on a 0-order AnYa type fuzzy rule-based system. In addition to the offline approach, we will also use the online version of the model, thereby allowing the hybrid model proposed in this paper to evolve on new stream samples to account for possible changes (drifts).

The following basic definitions of EDA will be referenced throughout the paper:-**x** = {$x_{1},x_{2},\;\ldots ,\;x_{k}$ } $\in R^{d}$: input variables. (where the index *k* indicates the time instance at which the data point arrives.)-**u** = {$u_{1}$, $u_{2}$, ..., $u_{L}$} $\in R^{d}$: set of unique data point locations.-$f_{1},f_{2},\;\ldots ,\;f_{L}$: number of times that different data points occupy the same unique locations.-$K^{c}$: the number of data samples ${\left\{x\right\}}_{K^{c}}^{c}$ belonging to the *c*-th class.-$U_{K}^{c}$: the number of unique data samples belonging to the *c*-th class.-$\sum_{c=1}^{C}K^{c}=K$ and $\sum_{c=1}^{C}U_{K}^{c}=U_{K}$.

Based on $u_{1}$, $u_{2}$, ..., $u_{L}$ and $f_{1},f_{2},\;\ldots ,\;f_{L}$, it is possible to reconstruct the dataset $x_{1},x_{2},\;\ldots ,\;x_{k}$ exactly if necessary, regardless of the order of arrival of the data points [12].

The SOF technique uses relevant concepts such as cumulative proximity, an empirical derivation of the model’s data without prior knowledge or previous assumptions being seen as a square form of distance. It can be defined by [29]:(2)$$\begin{array}{cc} \pi_{K}\left(x_{i}\right)=\sum_{j=1}^{K}d^{2}\left(x_{i},x_{j}\right); & i=1,2,\;\ldots ,\;K \end{array}$$
where d($x_{i},x_{j}$) denotes the Euclidean distance between $x_{i}$ and $x_{j}$ [29].

Another fundamental factor for the determination of data clouds is the unimodal density, which in turn is defined by [29]:(3)$$D_{K}\left(x_{i}\right)=\frac{\sum_{l=1}^{K}\pi_{K}\left(x_{l}\right)}{2K\pi_{K}\left(x_{i}\right)}=\frac{\sum_{l=1}^{K}\sum_{j=1}^{K}d^{2}\left(x_{l},x_{j}\right)}{2K\sum_{j=1}^{K}d^{2}\left(x_{i},x_{j}\right)};\;i=1,2,\;\ldots ,\;K$$

Finally, the third EDA operator relevant to the fuzzification approach used in this paper is the multimodal density, which can be estimated through a unique data sample, $u_{i}$, through the weighted sum of its unimodal density by its repetition times. It is calculated using [29]:(4)$$D_{K}^{MM}\left(u_{i}\right)=f_{i}D_{K}\left(u_{i}\right)=f_{i}\frac{\sum_{l=1}^{K}\pi_{K}\left(x_{l}\right)}{2K\pi_{K}\left(u_{i}\right)};\;i=1,2,\;\ldots ,\;U_{K}$$

These three EDA operators $\pi_{K}\left(x_{i}\right),D_{K}\left(x_{i}\right)$ and $D_{K}^{MM}\left(u_{i}\right)$ are fundamental to the recursive procedures to be performed by the fuzzification approach. They allow models that use the technique to act dynamically with data in the stream. The values of the EDA elements listed above can be updated using different types of distances/dissimilarity. In this paper, we will use the cosine distance, as suggested in Gu and Angelov [29]. Ways of updating the parameters recursively can be found in Angelov [12].

The (online) update and evolving procedure takes place by category, where SOF can identify each class’s prototypes separately and form a fuzzy rule of type AnYa of order 0 based on the prototypes identified per class. These prototypes are identified based on the teaching and mutual distributions of the data samples, through calculating the multimodal density in all the unique data samples using the following equation [29]:(5)$$D_{K^{c}}^{MM}\left(u_{i}^{c}\right)=f_{i}^{c}\frac{\sum_{l=1}^{K^{c}}\sum_{j=1}^{K^{c}}d^{2}\left(x_{l}^{c},x_{j}^{c}\right)}{2K^{c}\sum_{j=1}^{K^{c}}d^{2}\left(u_{i}^{c},x_{j}^{c}\right)},\;i=1,2,\;\ldots ,\;U_{K}^{c}$$
where *c* represents *c*-th class, *C*, ${\left\{x\right\}}_{K^{c}}^{c}=\left\{x_{1}^{c},x_{2}^{c},\;\ldots ,\;x_{K^{c}}^{c}\right\}{\left\{x\right\}}_{K^{c}}^{c}\subset {\left\{x\right\}}_{K}$ are the data samples of the *c*-th class and ${\left\{u\right\}}_{U_{K}^{c}}^{c}=\left\{u_{1}^{c},u_{2}^{c},\;\ldots ,\;u_{U_{K}^{c}}^{c}\right\}$ and ${\left\{f\right\}}_{U_{K}^{c}}^{c}=\left\{f_{1}^{c},f_{2}^{c},\;\ldots ,\;f_{U_{K}^{c}}^{c}\right\}$ correspond to a unique data sample set and frequencies of occurrence, respectively. $K^{c}$ is the number of data samples of the *c*-th class and $U_{K}^{c}$ is the number of unique data samples of the *c*-th class [29].

After this definition, a list **r** is used to organize previously calculated values, mainly between mutual distances and multimodal density values, defining as $r_{1}$ the element with the highest modal density. Therefore, the next elements, $r_{2},r_{3}\ldots r_{n}$, are calculated from the sample of data with the minimum distance to the previously indexed element. Therefore, $r_{2}$ is related to $r_{1}$, just as $r_{3}$ is related to the distance to $r_{2}$, and so on, until all samples are indexed [29].

After indexing the terms in the list **r**, the list of prototypes **p** is defined, which are then identified as the local maximum of the multimodal densities classified through the following condition [29]:(6)$$\begin{array}{cc} IF\;\left(D_{K^{c}}^{MM}\left(r_{i}\right)>D_{K^{c}}^{MM}\left(r_{i+1}\right)\right)\;AND\;\left(D_{K^{c}}^{MM}\left(r_{i}\right)>D_{K^{c}}^{MM}\left(r_{i-1}\right)\right) & THEN\;\left(r_{i}\in p\right) \end{array}$$

After determining all the prototypes, the relevant ones are discarded through a filtering process. This process begins with prototypes to attract samples of nearby data to form clouds of data. This operation can be expressed by [29]:(7)$$\begin{array}{cc} \Upsilon \;p=\underset{p_{n}\in p}{\arg\min}\left(d\left(x_{i},p_{n}\right)\right); & x_{i}\in {\left\{x\right\}}_{K^{c}}^{c} \end{array}$$
where Υp is the winning prototype. After the formation of all data clouds formed around the existing prototypes, the clouds’ centers $\varphi_{i}$ are formed. The multimodal values of the centers are also calculated and defined as follows [29]:(8)$$D_{K^{c}}^{MM}\left(\varphi_{i}\right)=S_{i}D_{K^{c}}\left(\varphi_{i}\right)$$
where $S_{i}$ is the support (number of members) of the *i*th data cloud. Then, for each data cloud, the collection of the centers of its neighboring data clouds is identified using the following principle [29]:(9)$$\begin{array}{cc} IF\;\left(d^{2}\left(\varphi_{i},\varphi_{j}\right)\leq G_{K^{c}}^{c,L}\right) & THEN\;\left(\varphi_{j}\in {\left\{\varphi \right\}}_{i}^{neighbouring}\right) \end{array}$$
where $\varphi_{j}\neq \varphi_{i}$ and $\;G_{K^{c}}^{c,L}$ the average radius of the influential local area around each data sample, which corresponds to the *L*th level of granularity and is derived from the data of the *c*-th class based on the user’s choice in an offline way [29]. Finally, the most representative prototypes of the *c*th class, denoted by $p_{c}$, are selected from the centers of the existing data clouds that satisfy the following rule [29]:(10)$$\begin{array}{cc} IF\;\left(D_{K^{c}}^{MM}\left(\varphi_{i}\right)>\max_{\varphi \in \varphi_{i}^{neighbouring}}\left(D_{K^{c}}^{MM}\left(\varphi \right)\right)\right) & THEN\;\left(\varphi_{i}\in p_{c}\right) \end{array}$$

Thus, the offline procedure of the data defines the centers that will reference the Gaussian projection that will be part of the first layer’s fuzzy neurons.

In each new sample submitted for evaluation in the online phase of the SOF, the SOF parameters are updated, allowing the approach to be consistent with the data streaming evaluation. Recursive updating allows parameters to be modified as new data samples come in and are presented to the training procedure. In this stage, the average radius of the local areas of influence is subsequently updated recursively. In online training, considering the new K + 1th instance, where a new data sample of the *c*-th class is represented by $x_{K^{c}+1}^{c}$, the parameters $\mu_{K^{c}+1}^{c},X_{K^{c}+1}^{c}$, and $\;\Sigma_{K^{c}+1}^{c}$ of the *c*-th class will be updated recursively by [29]:(11)$$\begin{array}{cc} \mu_{K^{c}+1}^{c}=\frac{K^{c}}{K^{c}+1}\mu_{K^{c}}^{c}+\frac{1}{K^{c}+1}x_{K^{c}+1}^{c}; & \mu_{1}^{c}=x_{1}^{c} \end{array}$$
(12)$$\begin{array}{cc} X_{K^{c}+1}^{c}=\frac{K^{c}}{K^{c}+1}X_{K^{c}}^{c}+\frac{1}{K^{c}+1}x_{K^{c}+1}^{cT}x_{K^{c}+1}^{c}; & X_{1}^{c}=x_{1}^{cT}x_{1}^{c} \end{array}$$
(13)$$\Sigma_{K^{c}+1}^{c}=\frac{K^{c}+1}{K^{c}}\left(X_{K^{c}+1}^{c}-\mu_{K^{c}+1}^{cT}\mu_{K^{c}+1}^{c}\right)$$

Please note that the centers and covariance matrices are updated class-wise independently, to achieve class-layered cloud partitions with $N^{c}$ prototypes for the *c*-th class.

In the recursive update process, the average radius of local areas of influence ($G_{K^{c}}^{c,L}$) (as used in Equation (Equation 9)) is updated a posteriori based on the ratio between $d_{K^{c}}^{c}$ and $G_{K^{c}}^{c,L}$ using:(14)$$G_{K^{c}+1}^{c,L}=\frac{d_{K^{c}+1}^{c}}{d_{K^{c}}^{c}}G_{K^{c}}^{c,L}=\frac{\frac{1}{{\left(K^{c}+1\right)}^{2}}\sum_{l=1}^{K^{c}+1}\pi_{K^{c}+1}\left(x_{l}^{c}\right)}{\frac{1}{{\left(K^{c}\right)}^{2}}\sum_{l=1}^{K^{c}}\pi_{K^{c}}\left(x_{l}^{c}\right)}G_{K^{c}}^{c,L}$$

The new sample evaluated by the SOF allows the approach to verify whether it should be come a new prototype. To do so, it must meet the following condition [29]:(15)$$\begin{array}{c} IF\;\left(D_{K^{c}+1}\left(x_{K^{c}+1}^{c}\right)>\max_{p\in {\left\{p\right\}}^{c}}\left(D_{K^{c}+1}\left(p\right)\right)\right)\;OR\;\left(D_{K^{c}+1}\left(x_{K^{c}+1}^{c}\right)<\min_{p\in {\left\{p\right\}}^{c}}\left(D_{K^{c}+1}\left(p\right)\right)\right)\\ THEN\;\left(x_{K^{c}+1}^{c}\mathrm{becomes}\;a\;\mathrm{new}\;\mathrm{prototype}\right) \end{array}$$

If Equation (Equation 15) is not fulfilled, another condition is evaluated [29]:(16)$$\begin{array}{cc} IF\;\left(\min_{p\in {\left\{p\right\}}^{c}}\left(d^{2}\left(x_{K^{c}+1}^{c},p\right)\right)>G_{K^{c}+1}^{c,L}\right) & THEN\;\left(x_{K^{c}+1}^{c}\;\mathrm{becomes}\;a\;\mathrm{new}\;\mathrm{prototype}\right) \end{array}$$

If the result is positive for this condition, a new prototype is added to the fuzzy rule of the *c*-th class, and the corresponding new data cloud with meta-parameters initialized by Equation (Equation 17) is added to the SOF classifier [29].
(17)$$\begin{array}{ccc} N^{c}\leftarrow N^{c}+1; & p_{N^{c}}^{c}\leftarrow x_{K^{c}+1}^{c}; & \begin{array}{cc} S_{N^{c}}^{c}\leftarrow 1; & {\left\{p\right\}}^{c}\leftarrow {\left\{p\right\}}^{c}+p_{N^{c}}^{c} \end{array} \end{array}$$
where $N^{c}$ = number of clusters/clouds. Otherwise, it will be assigned to the nearest prototype, and the corresponding data cloud meta-parameters are updated as follows [29]:(18)$$\begin{array}{cc} p_{n*}^{c}\leftarrow \frac{S_{n*}^{c}}{S_{n*}^{c}+1}p_{n*}^{c}+\frac{1}{S_{n*}^{c}+1}x_{K^{c}+1}^{c}; & S_{n*}^{c}\leftarrow S_{n*}^{c}+1 \end{array}$$
with $p_{n*}^{c}$ being the nearest prototype to $x_{K^{c}+1}^{c}$ in the *c*-th class layer and $S_{n*}^{c}$ its support (the number of past samples belonging to it).

For each cloud built by SOF, a Gaussian neuron is created. In this case, the choice of a nonparametric approach determined by the density of the problem data, allows the evolving fuzzy neural network architecture to be optimized to resolve the proposed problem. Figure 2 shows the global density of two features of the heart sounds problem.

#### 2.1.2. Weight Definition: Leave-One-Feature-Out Criterion

The first layer Gaussian neurons are formed with the centers and the SOF algorithm’s sigma value. These values allow the construction of the representation of the data in a fuzzified way. For the neuron’s composition, random values in the range between 0 and 1 are commonly used in fuzzy neural network approaches. However, this can impair the model’s interpretability. Therefore, we propose an alternative way, that is, to extract feature weights, which indicate their discriminatory power among the classes and thus their importance for the learning problem and interpretation of the relationships between the input space and the classes.

The central approach is to extract the feature weights based on the use of a class separability criterion, which in Lughofer [13] is established based on a robust expansion of Fisher’s separability criterion [30]. This criterion can be defined by:(19)$$J=trace(S_{w}^{-1}S_{b})$$
where $trace(S_{w}^{-1}S_{b})$ defines the sum of the diagonal elements of the matrix $S_{w}^{-1}S_{b}$ [13]. $S_{b}$ expresses the dispersion matrix between classes that measures the class’s dispersion averages to the total average, and $S_{w}$ denotes the within scattering matrix that measures the samples’ dispersion in their class averages. The latter can be determined through the average of the class-wise covariance matrices for exact formulas; see [13]. The higher *J* becomes, the better the separability of the classes.

Our approach is then to use the criterion in Equation (Equation 19) for determining the weight of each feature for discriminating between the classes. Thereby, we employ the leave-one-feature-out (LOFO) approach, which discards each feature from the complete set and then calculates Equation (Equation 19) for each obtained subset → obtaining $J_{1},\;\ldots ,\;J_{N}$ for *N* features in the dataset. This means that a higher drop in Equation (Equation 19) (and thus a lower value of $J_{i}$) indicates that feature *i* is more important as it was left out. Thus, and aiming for a relative importance among all features (relative to the most important feature, which should receive a weight of 1), the feature weights are assigned by:(20)$$w_{j}=1-\frac{J_{j}-\min_{i\;=\;1,\;\ldots ,\;N}J_{i}}{\max_{i\;=\;1,\;\ldots ,\;N}J_{i}}$$

That guarantees that the feature causing the minimal value of $J_{j}$ (hence most important one) gets a weight of 1, whereas all others are relatively compared to the minimum: when they are close, the weight will also be almost 1; in the case when they are significantly higher, the weight will be forced towards the ratio between minimum and maximum.

The separability criterion’s incremental update acts on the matrices represented in $S_{w}$ and $S_{b}$. These matrices can be updated during the incremental mode uniquely. In particular, $S_{b}$ can be updated by updating the class-wise and the overall mean through an incremental mean formula. Recursive covariance update formulas update the covariance matrices per class using rank-1 modification for more robust and faster convergence to the batch calculation [31]; see [13] for further details.

This approach allows the neuron’s weight that represents the problem’s feature as consistent with the nature of the data evaluated. If this dimension is hugely relevant to solving classifying patterns, that neuron will have a very expressive value (close to 1). The dimension that is less relevant to the differentiation of the classes of the problem, on the other hand, will have its value closer to zero. This factor presented facilitates the construction of better interpretable rules, as features with low weights can be discarded from the antecedents when showing the rules to experts (reduction of rule length, better readability). Figure 3 demonstrates the technique presented in the dataset used in this paper. It is possible to identify that MFCC 4 and MFCC 3 features are the most relevant features of this problem, according to the LOOF technique.

### 2.2. Second Layer—Logical Neuron

The second layer of the model consists of type III fuzzy neurons (that is, those representing the data through fuzzy rules) responsible for aggregating the neurons of the first layer and their respective weights. It is composed of *L* fuzzy logical neurons that perform a weighted aggregation of all of the first layer neurons using fuzzy operators called t-norm and t-conorm. This aggregation is performed using the feature weights $w_{il}$ (for *i* = 1 ...*N* and *l* = 1 ...*L*) defined in the range of 0 to 1. For each input variable *j*, only one first layer output $a_{jl}$ is defined as input of the *l*th neuron, so that **w** is sparse, and each neuron of the second layer is associated with an input variable.

The logical neurons can be of the AND-neuron, OR-neuron, UNI-neuron, NULL-neuron or UNINULL-neuron type. These neurons are built using fuzzy operators that can use the t-norm or t-conorm to complete their calculations. One neuron differs from the other because of its operator and the ability to express fuzzy rules differently. The following table assists in understanding the terms.

*S* and *s* are t-conorm; *T* and *t* are t-norm; (*g*) = identity element; (*u*) = absorption element; $U_{1}$ is a uninorm with an identity element = $\frac{g}{\beta }$; and $U_{2}$ is a uninorm with ($\frac{u-\beta }{1-\beta }$) as an identity element.

The great advantage of fuzzy logic operators is that they can sometimes use t-norms, sometimes t-conorms, allowing the neuron to perform operations like AND-neuron or OR-neuron, depending on the context. The function p (responsible for transforming the inputs and corresponding weights into individual transformed values), when applicable to neurons, performs the following steps:1Each pair ($a_{i}$, $w_{i},\;\ldots$) is converted into one value $b_{i}$ = *p**($a_{i}$, $w_{i}$)*;2Calculate the aggregation of all values using the fuzzy operators (U, NU or $N^{un}$) *($b_{1},b_{2}\ldots b_{n}$)*, where *n* is the number of inputs.

Therefore, this paper’s fuzzy rules to solve the problem of cardiac sounds may have connective AND or OR, depending on the type of neuron in the model structure.

### 2.3. Third Layer

The third layer of the model is represented by a neural aggregation network responsible for the defuzzification procedure. Therefore, it receives the weights of the second layer’s rules and performs the proper calculations to obtain the desired outputs.

The output layer is constituted of one neuron (which can be recognized as a singleton) whose activation function is established by:(21)$$\begin{array}{c} y=sign\sum_{j=0}^{L}f_{\omega }\left({\vec{Z}}_{rul_{j}},{\vec{v}}_{j}\right) \end{array}$$
where ${\vec{Z}}_{rul_{0}}$ = 1; ${\vec{v}}_{0}$ is the bias; ${\vec{Z}}_{rul_{j}}$ and ${\vec{v}}_{j}$, *j* = 1, ..., *L* are the outputs of the fuzzy neurons of the second layer and their corresponding weights, respectively; $f_{\omega }$ is the activation function (linear); and sign represents the signal function.

The signal function can be represented by:(22)$$sign=\left\{\begin{array}{cc} -1 & ,\mathrm{if}\;\sum_{j\;=\;0}^{L}f_{\omega }\left({\vec{Z}}_{rul_{j}},\;{\vec{v}}_{j}\right)<0\\ 1 & ,\mathrm{if}\;\sum_{j\;=\;0}^{L}f_{\omega }\left({\vec{Z}}_{rul_{j}},\;{\vec{v}}_{j}\right)>0 \end{array}\right.,$$

The signal function is necessary for the treatment of the model outputs in binary classification, as is the problem to be addressed in this paper.

### 2.4. Recursive Estimation of Output Weights

The defuzzification process involves turning fuzzy rules into numerical solutions to the problem. In the case of the hybrid model proposed in this paper, the definition of the weights helps in the process of interpretability of the fuzzy rules, and at the same time, helps the neural aggregation network to classify the samples of the cardiac sounds problem. Therefore, the initial weights for the artificial neurons are estimated from the initial dataset through the Moore–Penrose pseudo-inversion technique, as also used in de Campos Souza [25], for binary classification problems. This leads to the estimation of a vector ${\vec{v}}_{k}$ per class separately through:(23)$${\vec{v}}_{k}={\vec{Z_{rul}}}^{+}{\vec{y}}_{k}\;\forall k=1,\;\ldots ,\;c$$
where ${\vec{Z_{rul}}}^{+}={\vec{Z}}_{rul}^{T}{\vec{Z}}_{rul}$ is the pseudo-inverse of the Moore–Penrose matrix [32] of ${\vec{Z}}_{rul}$. ${\vec{Z}}_{rul}$ contains the activation levels (as rows) for all selected neurons (as columns). $y_{k}$ denotes the column indicator vector containing 1 s at the row positions for samples belonging to class k and 0 for all samples belonging not to class *k*. This leads to an indicator-based least squares estimation (regression problem on {0,1}), where a different output weight vector ${\vec{v}}_{k}$ (over all selected neurons) per class is achieved. That is, following the well-known one-versus-rest classification realized through regression by indicator matrix [33], which we extend here to a nonlinear version according to the integration of the activation levels in the regression matrix *Z*. According to the analysis in [34], the nonlinear version does not suffer from any class masking effects (as is the case for the classical linear version; see [33]).

For new incoming online stream samples, the weights are recursively updated to meet the system dynamics. Therefore, we employ the recursive weighted least squares (RWLS) approach [35], which is a fast converging well-known estimator, as an incremental Gauss–Newton step is embedded, which can converge to the real optimum within single iterations (thus, single sample updates). We extend RWLS to an indicator-based version for each class, termed as I-RWLS, with formulas for updating the $\vec{v_{k}}$ column vector for the *k*th class given by (with an update from sample t−1 to *t*):(24)$$\eta ={\vec{z}}^{t}Q^{t-1}{\left(\lambda +{\left({\vec{z}}^{t}\right)}^{T}Q^{t-1}{\vec{z}}^{t}\right)}^{-1}$$
(25)$$Q^{t}=\left(I_{L^{t}}-\eta^{T}{\vec{z}}^{t}\right)\lambda^{-1}Q^{t-1}$$
(26)$${\vec{v}}_{k}^{t}={\vec{v}}_{k}^{t-1}+\eta^{T}\left(y_{k}^{t}-{\vec{z}}^{t}{\vec{v}}_{k}^{t-1}\right)$$
where the index *k* again denotes the class index k=1,…,c. ${\vec{z}}^{t}$ denotes the regressor (row) vector of the current sample (i.e., the activation levels of all neurons in the current stream sample), *J* is the current kalman gain (row) vector and $I_{L_{s}^{t}}$ is an identity matrix based on the number of neurons in the second layer, $L_{s}^{t}\times L_{s}^{t}$; λ∈]0,1] denotes a possible forgetting factor, but is to 1 per default (no forgetting). *Q* denotes the inverse Hessian matrix $Q={\left({\vec{Z}}_{rul}^{T}{\vec{Z}}_{rul}\right)}^{-1}$ and is set initially as $\omega I_{L_{s}^{t}}$, where ω is a big number (e.g., 1000), such as in Rosa et al. [36]; see also Chapter 2 in [37] for a detailed convergence analysis. This matrix is directly and incrementally updated by the second equation above without requiring (time-consuming and possibly unpredictable) re-inversion of matrices. ψ is a forgetting for increased flexibility during learning (thus, being able to address concept drifts), but is set to the default value of 1 (no forgetting) in all our test experiments. The training algorithm proposed in this paper can be viewed below (Algorithm 1):
**Algorithm 1** EFNN-SOF training and update algorithm.**Initial Batch Learning Phase (Input: data matrix *X* with *K* samples):**  1:Extract *L* clouds in the first layer using the SOF approach (*L* is automatically estimated therein).  2:Estimate center values $\vec{c}$ and widths $\vec{\sigma }$ for the *L* clouds derived from SOF.  3:Calculate the combination (feature) weights $\vec{w}$ for neuron construction using Equation (Equation 20).  4:Construct *L* logic neurons on the second layer of the network by welding the *L* fuzzy neurons of the first layer, using logical neurons concept and weights $\vec{w}$.  5:   6:**for**i=1,…,K**do**  7:  Calculate the regression vector $z({\vec{x}}_{i})$ by activation levels of all neurons in ${\vec{x}}_{i}$.  8:  Store it as one row entry into the activation level matrix *Z*.  9:**end for**10:Extract reduced activation level matrix $Z_{rul}$ according to the $L_{s}<L$ selected neurons.11:Estimate the weights ${\vec{v}}_{k}$ of the output layer for all classes k=1,…,c by Equation (Equation 23) using $Z_{rul}$ and indicator vectors ${\vec{y}}_{k}$.**Update Phase (Input: single data sample ${\vec{x}}_{t}$):**  1:Update $L_{s}$ clouds and evolving new ones on demand in the first layer using evolving SOF approach (→$L_{s,upd}$ clouds).   2:Update the feature weights $\vec{w}$ by updating the within- and between-class scatter matrix and recalculating Equation (Equation 20).  3:Perform Steps 2 and 4 with $L_{s,upd}$ clouds.  4:Calculate the degree of change of all neurons (rules) by Equation (Equation 29).  5:Calculate the regression vector $z({\vec{x}}_{t})$ by activation levels of all neurons in ${\vec{x}}_{t}$.  6:Update the weights ${\vec{v}}_{k}$ of the output layer by Equation (Equation 26).

### 2.5. Interpretability Through Fuzzy Rules

The interpretability of hybrid models becomes essential for artificial intelligence to stop being something incomprehensible and for its results to help companies and people to understand the reason for their results. Fuzzy neural networks work with building fuzzy rules. In this paper, they can be expressed by: (27)$$\begin{array}{c} Rule_{1}:\;If\;x_{1}\;is\;A_{1}^{1}\;with\;impact\;w_{11}\ldots \\ and/or\;x_{2}\;is\;A_{1}^{2}\;with\;impact\;w_{21}\ldots \\ and/or\;x_{n}\;is\;A_{1}^{n}\;with\;impact\;w_{n1}\ldots \\ Then\;y_{1}\;is\;v_{1}\\ Rule_{L}:\;If\;x_{1}\;is\;A_{L}^{1}\;with\;impact\;w_{1L}\ldots \\ and/or\;x_{2}\;is\;A_{L}^{2}\;with\;impact\;w_{2L}\ldots \\ and/or\;x_{n}\;is\;A_{L}^{n}\;with\;impact\;w_{nL}\ldots \\ Then\;y_{L}\;is\;v_{L} \end{array}$$
with $A_{i}^{1},\;\ldots ,\;A_{i}^{n}$ fuzzy sets represented as linguistic terms for the *n* inputs appearing in the *i*-th rules, and $y_{1},\;\ldots ,\;y_{L}$ are consequent terms, in our case classification responses (either 0 = normal heart sounds or 1 = heart murmur); thus, each rule stands for one concrete readable relationship between the input features and the presence of heart murmur; whenever a weight $w_{ij}$ is close to 0, the corresponding antecedent part can be neglected when showing the rules to experts; this reduces the rule length and increases the transparency and readability of the rules.

In the case of using UNINULL-neuron, the fuzzy rules are represented as follows:(28)$$\begin{array}{c} Rule_{1}:\;Ifx_{i1}\;is\;A_{1}^{1}\;with\;impact\;w_{11}\ldots \\ and/or_{\left(g,u,\beta \right)}\;x_{i2}\;is\;A_{1}^{2}\;with\;impact\;w_{21}\ldots \\ and/or_{\left(g,u,\beta \right)}\;x_{iN}\;is\;A_{1}^{N}\;with\;impact\;w_{N1}\ldots \\ Then\;y_{1}\;is\;v_{1}\\ Rule_{2}:\;Ifx_{i1}\;is\;A_{M}^{1}\;with\;impact\;w_{1L}\ldots \\ and/or_{\left(g,u,\beta \right)}\;x_{i2}\;is\;A_{M}^{2}\;with\;impact\;w_{2L}\ldots \\ and/or_{\left(g,u,\beta \right)}\;x_{i2}\;is\;A_{M}^{N}\;with\;impact\;w_{NL}\ldots \\ Then\;y_{L}\;is\;v_{L} \end{array}$$

The triple values involved in these rules allow the fuzzy neuron to act with different operators. Table 1 explains the relationship between g, u and β and the respective fuzzy operators.

The complexity of these rules is linked to the number of dimensions of the problem. Therefore, any technique that can identify the relevance of a feature to a problem becomes fundamental for reducing the complexity of the rules, because when the impact (defined by the weight of the Gaussian neuron) is small, that dimension can be disregarded from the rule generated. Factors like this and updating rules through evolving training, facilitate the construction of knowledge about the problem. The antecedent connectors can be of the AND or OR type, as they can vary according to the type of neurons used in the second layer.

In the case of binary problems, the weights of the rules indicate that the rule is closer to indicating normal behavior the lower its weight for the problem, in the same way that high values of *v* indicate that the rule contributes enormously to the identification of heart murmur through the analysis of cardiac sounds.

#### Degree of Rule Variations over Time

Aside from the linguistically readable if–then structure, which can be directly extracted from the produced clouds, as described in Section 2, a different form of interpretability in data stream modeling (evolving sense) would be to track the degree of modification to the rules over time. That may be a crucial indicator within an interactive background for users/experts to inform them whether and when to re-inspect changed rules: only in the occurrence of notable variations is it worth doing so. With this insight, the degree of change of rules could be used as a structural active learning based trigger for demanding expert feedback, indicating whether the knowledge of rules has significantly improved during the adaptation of new data samples.

To meet online and especially single-pass demands, we suggest using a fast geometric-based criterion to measure the degree of change of a rule, which does not expect any past data samples. Consequently, we lean on the idea in Lughofer [38], where the intersection degree of Gaussian membership functions is used as an indicator of whether two distinctive rules have become overlapping, which can occur over data stream samples due to cluster fusion impacts. In particular, consider investigating the overlap degree of a rule *i* before and after its update (termed as $R_{i}\left(bef\right)$ and $R_{i}\left(after\right)$) dimension-wise and calculating an amalgamated value that can be applied as a similarity degree $S(R_{i}\left(bef\right),R_{i}\left(after\right))$. The degree of change is then given by
(29)$$\Lambda \left(R_{i}\right)=1-S\left(R_{i}\left(bef\right),R_{i}\left(after\right)\right)$$
bef=N−n and after=N assuming that *n* new samples have passed the data stream-based adaptation phase with *N* samples used so far for model training and evolution.

The dimension-wise calculation of *S* is justified because the antecedents of fuzzy rules are regularly represented by a conjunction of single dimension-wise membership functions. Therefore, the approach is per the viewpoint of a fuzzy rule base: two rules are only similar if all their antecedent parts are similar. The x-coordinates of the intersection points of two Gaussians used as fuzzy sets in the same antecedent part of rule *i* (here for the *j*-th) before and after its update can be calculated by [38]:(30)$$\begin{array}{cc} inter_{x}\left(1\right) & =-\frac{c_{bef,j}\sigma_{after,j}^{2}-c_{after,j}\sigma_{bef,j}^{2}}{\sigma_{bef,j}^{2}-\sigma_{after,j}^{2}}\\ \; & +\sqrt{{\left(\frac{c_{bef,j}\sigma_{bef,j}^{2}-c_{bef,j}\sigma_{after,j}^{2}}{\sigma_{after,j}^{2}-\sigma_{bef,j}^{2}}\right)}^{2}-\frac{c_{bef,j}^{2}\sigma_{after,j}^{2}-c_{after,j}^{2}\sigma_{bef,j}^{2}}{\sigma_{after,j}^{2}-\sigma_{bef,j}^{2}}}\\ inter_{x}\left(2\right) & =-\frac{c_{bef,j}\sigma_{after,j}^{2}-c_{after,j}\sigma_{bef,j}^{2}}{\sigma_{bef,j}^{2}-\sigma_{after,j}^{2}}\\ \; & -\sqrt{{\left(\frac{c_{bef,j}\sigma_{bef,j}^{2}-c_{bef,j}\sigma_{after,j}^{2}}{\sigma_{after,j}^{2}-\sigma_{bef,j}^{2}}\right)}^{2}-\frac{c_{bef,j}^{2}\sigma_{after,j}^{2}-c_{after,j}^{2}\sigma_{bef,j}^{2}}{\sigma_{after,j}^{2}-\sigma_{bef,j}^{2}}} \end{array}$$
with being $c_{bef,j}$ the *j*-th center coordinate of the rule before its update, and $c_{after,j}$ the *j*-th center coordinate of the rule after its update.

The maximal membership degree of the two Gaussian membership functions in the intersection coordinates is then used as the overlap and thus similarity degree of the corresponding rules’ antecedent parts in the *j*-th dimension: (31)$$S_{bef,after}\left(j\right)=\max\left(\mu_{i}\left(inter_{x}\left(1\right)\right),\mu_{i}\left(inter_{x}\left(2\right)\right)\right)$$
with being $\mu_{i}\left(inter_{x}\left(1\right)\right)$ the membership degree of the *j*-th fuzzy set in rule *i* in the intersection point $inter_{x}\left(1\right)$ (which is the same before and after the update due to intersection). The amalgamation overall rule’s antecedent parts leads to the final similarity degree between the rule before and after its update:(32)$$S\left(R_{i}\left(bef\right),R_{i}\left(after\right)\right)=Agg_{j=1}^{p}S_{bef,after}\left(j\right)$$
where Agg denotes an aggregation operator and *p* is the number of inputs. A feasible choice is a t-norm [39], as a strong non-overlap along one single dimension is sufficient that the clusters do not overlap at all (hence are dissimilar, thus *S* should become a low value); this is because one dimension can already be sufficient for tearing the rules away in a geometric sense; see also [38].

## 3. Materials and Methods

The dataset used in this study came from a set of public data provided by Liu et al. [4]. It is the combination of three datasets. The University of Michigan Health System provided the first dataset with 23 recordings of heart sounds with a total duration of 1496.8 s. The second dataset that makes up the data used in this research is also published, comprising 176 recordings for segmentation of cardiac sounds and 656 recordings for cardiac sound classification lasting from 1 to 30 s and having a frequency range of 195 Hz. Finally, 64 final recordings were from the non-open database of cardiac auscultation of heart murmurs, provided by eGeneral Medical Inc. All three bases were combined and were the reference of studies by Liu et al. [4]. At the PhysioNet/CinC Challenge 2016, an extensive collection of heart sound recordings was added to the dataset. These data came from different clinical and non-clinical environments in the real world. The data include not only clear heart sounds but also very loud recordings, providing authenticity to the challenge. The data were collected from normal individuals and people with some cardiac problems—mostly children and adults—from different locations on the patients’ bodies, with a predominance of the aortic area, pulmonary area, tricuspid area and mitral area [4]. These data were collected to construct the dataset through the union of large databases that used different equipment to collect them. For data collected by the Massachusetts Institute of Technology (MIT), an electrocardiogram (ECG) with an electronic stethoscope Welch Allyn Meditron (Skaneateles Falls, New York, NY, USA) was used, being capable of a frequency response between 20 and 20 kHz. The Aalborg University database used a Littmann E4000 electronic stethoscope (3M, Maplewood, Minnesota) with the stethoscope’s frequency response capacity ranging between 20 and 1000 Hz. The Aristotle University of Thessaloniki used the audio scope (electronic stethoscope) in order to record amplified and unfiltered signals. In the KN Toosi University of Technology database, an electronic stethoscope (3M Littmann’s 3200) in conjunction with echocardiography was used to collect the data. In the work carried out by the University of Haute Alsace, the stethoscopes provided by Infral Corporation (Strasbourg, France) were used. Finally, Dalian University of Technology and Shiraz University used electronic stethoscopes from different companies.

Table 2 summarizes the main dimensions, average values and their respective standard deviations, in addition to maximum and minimum values for each feature.

Figure 4 shows the organization of the dataset features evaluated in this paper across all samples. In this figure, it is possible to identify the complexity of the data.

The procedures to be performed with this type of dataset involve tasks such as recording sounds in selected patients, pre-processing the collected signals and segmenting and classifying the collected data so that medical procedures can be performed in cases of any cardiac problem detected. In the pre-processing stage of the signal, it is interesting to highlight the selection of the best characteristics in the collected signals and the application of filters to detect baseline changes and high-frequency noise. Thus, the data will be ready to be delineated as to the beginning and end of a heartbeat in different elements (S1, S2, systolic and diastolic). This identification occurs in the segmentation phase so that in the end, in the classification phase, the patterns of cardiac signals are determined and compared with signals from cardiac pathologies. In this case, intelligent systems based on artificial intelligence can continuously act to classify these heart-related disease patterns. After defining an abnormal pattern or that corresponds to known heart disease, medical procedures and treatments can be performed [4]. The use of intelligent models facilitates and automates the decision-making process. Therefore, it is essential to expand the studies related to heart murmurs’ identification by approaches based on artificial intelligence.

The results obtained by related works depend on techniques for feature selections or complementary intelligent approaches to obtain these results. In this paper, all dimensions of the problem will be used to obtain interpretable assessments on the heart murmur problem. Another relevant factor in the results obtained by RNN and HMM can be explained due to benefiting from its remarkable ability to model sequential patterns because the data in this article are essentially sequences. It is worth mentioning that the approaches also worked offline or with a lower percentage of training or testing, diverging from the evolving approaches proposed in the experiments.

## 4. Heart Sounds Classification Test

This section presents the tests to be performed on the dataset—the models’ configurations, the hyperparameters, forms of evaluation and the tests’ results. The tests were configured based on a specific proportion of samples for offline training (10%), and the rest were used for online adaptive training to prove the efficiency of the model’s evolving approach. The tests were compared with the models proposed in this paper, where each model constructed a different type of neuron according to the variants shown in Table 3. Therefore, it is identified as “evolving-SOF-type” in the legend of the figures and in the text. For example, a model that uses AND-neurons is represented in the graphics as evolving-SOF-AND. These models were compared with a renowned state-of-the-art technique, termed ALMMo-0* [10], for the online detection of cardiac sounds. The parameters used by the ALMMo-0* model were the same as those suggested and successfully used in Soares et al. [10]. To help compare the results of the online models; the model in its offline version of the fuzzification process was also used in the tests.

Simulations were performed on a Core (TM) 2 Duo CPU, 2.27 GHz, with 3 GB RAM. Time is represented by the sum of training time and test (seconds) in each of the models. Neurons represent the most representative neurons after the pruning or regularization of the models.

### 4.1. Evaluation Criteria

The factors evaluated in this paper are based on the accuracy, calculated in batch offline mode as:(33)$$Accuracy=\frac{TP+TN}{TP+FN+TN+FP}$$
where TP= true positive, TN= true negative, FN= false negative and FP= false positive. For the online mode we exploit the usage of accuracy trend lines over time in order to see the evolution of the accuracy, as a result of the evolution of the models. Such an accuracy trend line can be achieved by the interleaved predict and then train evaluation scenario [43]. In this scenario, (i) a prediction is made on a new incoming stream sample; (ii) the prediction $\hat{y}$ is compared with the real class label *y* (always included in our evaluation dataset); (iii) the accuracy is updated according to the correctness of the prediction by:(34)$$Accuracy\left(K+1\right)=\frac{Accuracy\left(K\right)*K+I_{\hat{y}=y}}{K+1},$$
where *I* denotes the indicator function and is 1 when the prediction is correct, i.e., $\hat{y}=y$; 0 otherwise (Accuracy(0)=0); and (iv) afterwards the model is updated. This means that the update of the accuracy in Equation (Equation 34) results in an accumulated one-step-ahead prediction accuracy, which can be stored in a vector for each *K* leading to an accuracy trend line—it is a commonly used measure for timely-tracked model accuracy in the stream mining community [44].

### 4.2. Results

In Figure 5, the accumulated accuracy over time is plotted in one graph for all methods (construction variants and SoA ALMMo-0*)—the closer the trend to 100% on the y-axis, the better the model performance. It is possible to notice that the models had different performances. It should be noted that the models with neurons of the UNI-neuron, NULL-neuron and UNINULL-neuron type obtained the best results of the experiment with similar behavior to each other. All three models have surpassed the state-of-the-art for resolving the identification of cardiac sounds in online training. It is important to note that models that used only AND-neurons and OR-neurons (the classical construction in fuzzy neural network approaches) were the worst performers, nor could they outperform the related approach ALMMo-0*. This factor can be explained due to the complex nature of the problem, which requires variations in the neuron constructions (allowing a combination of ANDs and ORs in rules) to efficiently solve the problem. The three models with such an advanced construction turned out to be the best performers, even being able to outperform ALMMo-0*, and this more or less over the whole timeline of all stream samples (thus, in each point of time showing a superior overall performance).

### 4.3. Further Discussions

When comparing the results with those shown in Table 4, the best neuron construction variant of the model in this paper showed a better (final) accuracy than those reported in Potes et al., Zahibi et al., Whitaker et al., Li et al. and Noman et al., even though these were trained on the full dataset (comprising around 10.3 K samples) in a batch, offline training mode, thereby having the ability to “see” all the data and iterate over the whole dataset multiple times, which is convenient in neural network-based learning schemes (such as stochastic gradient descent approaches and others). This is remarkable performance on the part of our method, which always uses only one sample for model updating, thereby having a much narrower view on the data. Another factor to be highlighted is the difference between the online and offline versions in solving the problem (the offline versions marked by the last four lines in the legend of Figure 5). Offline models use the same fuzzy rules with the same parameters as initially learned to classify new patterns. Thus, they are hardly able to improve the accuracy over time as the evolving methods significantly do (up to more than 90% accuracy) due to the integration of new data in their parameter and structure updates.

It should be noted that the superior accuracy results obtained by those obtained in Dominguez-Morales et al. were also obtained with an offline model. In Latif et al., Zeng et al. and Aziz et al., the tests were conducted by explicit feature selection techniques before the final results. Therefore, not all the features of the problem were used and a kind of curse of dimensionality reduction could be achieved, but this was done in connection with external expert knowledge (which requires manual effort and may be not necessarily available). In the tests by Shukla et al., the accuracy results were obtained only with a subset of 3669 samples selected by different criteria. Chen et al.’s work also obtained better numerical results through the use of a modified frequency wavelet transform as a pre-processing step on the feature space. It should be noted that all these results from related works which were numerically superior to our model were not produced in an evolving way, but fully in an offline, batch training phase seeing the whole set at once. Such models, however, cannot be applied for a larger (possibly infinite) online training process, where stream samples come in one by one (and are not collected and stored as a full dataset in advance). Another critical factor is the lack of knowledge generation by state-of-the-art models on the theme of cardiac sounds. The authors emphasized the techniques and results obtained, but did not demonstrate the final relationship between the problem’s features and the output classes (heart murmur or no hear murmur), as their models appeared to be black boxes without any show of interpretable components (such as rules).

The fuzzy rules of the problem had as a starting parameter the total of four rules. After the evolving, adaptive training phase was entirely performed on all the stream samples, 49 fuzzy rules turned out to be generated to address the problem. That means that the model autonomously learned from the new samples received and created new logical interactions to solve the problem, which in turn led to a rising accumulated accuracy trend line over time, as shown in Figure 5. These characteristics are very similar to human behavior when learning new things and are then able to make better estimations/predictions when new situations arise. Regarding the initial fuzzy rules, a set of four partitions for each feature was used to represent knowledge (Table 5). Regarding each feature’s impact, it is possible to highlight that all features initially had high impact values in the fuzzy rules, where the most significant emphasis goes to MFCC 13. After the evolution training, the LOO technique adequately identified the weight of features for the problem, where the most relevant for identifying cardiac sounds were MFCC 3 and 4—as shown in Figure 3.

During the offline training step, four rules were created. For the interpretability criteria, the following conversions were considered: the first membership function is very small; the second membership function represents small; the third membership function represents medium; and finally, the fourth membership function is presented as high. Thus, the offline rules can be described and interpreted as follows:

Rule 1—If (mean value is small) with impact 0.9179 or (median value is small) with impact 0.9332 or (standard deviation is very small) with impact 0.9466 or (mean absolute deviation is small) with impact 0.9401 or (quantile 25 is small) with impact 0.9066 or (quantile 75 is small) with impact 0.9066 or (signal iqr is small) with impact 0.9066 or (sample skewness is very small) with impact 0.9064 or (sample Kurtosis is small) with impact 0.9290 or (signal entropy is very small) with impact 0.9078 or (spectral entropy is small) with impact 0.9372 or (dominant frequency value is small) with impact 0.9076 or (dominant frequency magnitude is small) with impact 0.9076 or (dominant frequency ratio is small) with impact 0.9261 or (MFCC 1 is small) with impact 0.9164 or (MFCC 2 is small) with impact 0.9071 or (MFCC 3 is small) with impact 0.9088 or (MFCC 4 is small) with impact 0.9159 or (MFCC 5 is small) with impact 0.9534 or (MFCC 6 is small) with impact 0.9068 or (MFCC 7 is small) with impact 0.9083 or (MFCC 8 is small) with impact 0.9101 or (MFCC 9 is small) with impact 0.9226 or (MFCC 10 is small) with impact 0.9361 or (MFCC 11 is small) with impact 0.9119 or (MFCC 12 is small) with impact 0.9085 or (MFCC 13 is small) with impact 1.0000 then (sound identification is normal heart sound) with certainty 0.2350.

Rule 2—If (mean value is medium) with impact 0.9179 or (median value is high) with impact 0.9332 or (standard deviation is high) with impact 0.9466 or (mean absolute deviation is high) with impact 0.9401 or (quantile 25 is high) with impact 0.9066 or (quantile 75 is high) with impact 0.9066 or (signal IQR is high) with impact 0.9066 or (sample skewness is high) with impact 0.9064 or (sample Kurtosis is high) with impact 0.9290 or (signal entropy is high) with impact 0.9078 or (spectral entropy is high) with impact 0.9372 or (dominant frequency value is high) with impact 0.9076 or (dominant frequency magnitude is high) with impact 0.9076 or (dominant frequency ratio is high) with impact 0.9261 or (MFCC 1 is high) with impact 0.9164 or (MFCC 2 is high) with impact 0.9071 or (MFCC 3 is very small) with impact 0.9088 or (MFCC 4 is medium) with impact 0.9159 or (MFCC 5 is high) with impact 0.9534 or (MFCC 6 is high) with impact 0.9068 or (MFCC 7 is high) with impact 0.9083 or (MFCC 8 is high) with impact 0.9101 or (MFCC 9 is high) with impact 0.9226 or (MFCC 10 is high) with impact 0.9361 or (MFCC 11 is high) with impact 0.9119 or (MFCC 12 is high) with impact 0.9085 or (MFCC 13 is high) with impact 1.0000 then (sound identification is heart murmur) with certainty −0.2782.

Rule 3—If (mean value is high) with impact 0.9179 and (median value is medium) with impact 0.9332 and (standard deviation is medium) with impact 0.9466 and (mean absolute deviation is medium) with impact 0.9401 and (quantile 25 is medium) with impact 0.9066 and (quantile 75 is medium) with impact 0.9066 and (signal IQR is medium) with impact 0.9066 and (sample skewness is small) with impact 0.9064 and (sample Kurtosis is medium) with impact 0.9290 and (signal entropy is medium) with impact 0.9078 and (spectral entropy is medium) with impact 0.9372 and (dominant frequency value is medium) with impact 0.9076 and (dominant frequency magnitude is medium) with impact 0.9076 and (dominant frequency ratio is medium) with impact 0.9261 and (MFCC 1 is small) with impact 0.9164 and (MFCC 2 is medium) with impact 0.9071 and (MFCC 3 is small) with impact 0.9088 and (MFCC 4 is high) with impact 0.9159 and (MFCC 5 is medium) with impact 0.9534 and (MFCC 6 is medium) with impact 0.9068 and (MFCC 7 is medium) with impact 0.9083 and (MFCC 8 is medium) with impact 0.9101 and (MFCC 9 is medium) with impact 0.9226 and (MFCC 10 is medium) with impact 0.9361 and (MFCC 11 is medium) with impact 0.9119 and (MFCC 12 is medium) with impact 0.9085 and (MFCC 13 is medium) with impact 1.0000 then (sound identification is normal heart sound) with certainty 0.3899.

Rule 4—If (mean value is very small) with impact 0.9179 or (median value is very small) with impact 0.9332 or (standard deviation is small) with impact 0.9466 or (mean absolute deviation is very small) with impact 0.9401 or (quantile 25 is very small) with impact 0.9066 or (quantile 75 is very small) with impact 0.9066 or (signal IQR is very small) with impact 0.9066 or (sample skewness is medium) with impact 0.9064 or (sample Kurtosis is very small) with impact 0.9290 or (signal entropy is small) with impact 0.9078 or (spectral entropy is very small) with impact 0.9372 or (dominant frequency value is very small) with impact 0.9076 or (dominant frequency magnitude is very small) with impact 0.9076 or (dominant frequency ratio is very small) with impact 0.9261 or (MFCC 1 is medium) with impact 0.9164 or (MFCC 2 is very small) with impact 0.9071 or (MFCC 3 is small) with impact 0.9088 or (MFCC 4 is very small) with impact 0.9159 or (MFCC 5 is very small) with impact 0.9534 or (MFCC 6 is very small) with impact 0.9068 or (MFCC 7 is very small) with impact 0.9083 or (MFCC 8 is very small) with impact 0.9101 or (MFCC 9 is very small) with impact 0.9226 or (MFCC 10 is very small) with impact 0.9361 or (MFCC 11 is very small) with impact 0.9119 or (MFCC 12 is very small) with impact 0.9085 or (MFCC 13 is very small) with impact 1.0000 then (sound identification is normal heart sound) with certainty 0.6577.

After evolution, 49 fuzzy rules were created, and their pertinence functions showed similar behaviors for the interpretation criteria. The following four rules represent the initial rules (as shown above) after their update with all stream samples. The rules’ consequent values are defined in the range between −1 and 1, where the first value indicates a strong probability of a heart murmur, and the value 1 represents a strong tendency to healthy heart sounds:

Rule 1—If (mean value is high) with impact 0.7192 or (median value is medium) with impact 0.7152 or (standard deviation is very small) with impact 0.7527 or (mean absolute deviation is very small) with impact 0.7257 or (quantile 25 is medium) with impact 0.7153 or (quantile 75 is small) with impact 0.7153 or (signal IQR is very small) with impact 0.7153 or (sample skewness is medium) with impact 0.7248 or (sample Kurtosis is small) with impact 0.7167 or (signal entropy is very small) with impact 0.7210 or (spectral entropy is small) with impact 0.7410 or (dominant frequency value is small) with impact 0.7283 or (dominant frequency magnitude is medium) with impact 0.7430 or (dominant frequency ratio is small) with impact 0.7210 or (MFCC 1 is small) with impact 0.7255 or (MFCC 2 is small) with impact 0.7170 or (MFCC 3 is small) with impact 0.7650 or (MFCC 4 is very small) with impact 1.0000 or (MFCC 5 is small) with impact 0.7429 or (MFCC 6 is very small) with impact 0.7157 or (MFCC 7 is small) with impact 0.7257 or (MFCC 8 is small) with impact 0.7170 or (MFCC 9 is small) with impact 0.7164 or (MFCC 10 is small) with impact 0.7196 or (MFCC 11 is small) with impact 0.7155 or (MFCC 12 is small) with impact 0.7218 or (MFCC 13 is very small) with impact 0.7326 then (sound identification is heart murmur) with certainty −1.

Rule 2—If (mean value is high) with impact 0.7192 or (median value is very high) with impact 0.7152 or (standard deviation is medium) with impact 0.7527 or (mean absolute deviation is medium) with impact 0.7257 or (quantile 25 is medium) with impact 0.7153 or (quantile 75 is high) with impact 0.7153 or (signal IQR is small) with impact 0.7153 or (sample skewness is medium) with impact 0.7248 or (sample Kurtosis is medium) with impact 0.7167 or (signal entropy is high) with impact 0.7210 or (spectral entropy is high) with impact 0.7410 or (dominant frequency value is very high) with impact 0.7283 or (dominant frequency magnitude is high) with impact 0.7430 or (dominant frequency ratio is very high) with impact 0.7210 or (MFCC 1 is high) with impact 0.7255 or (MFCC 2 is medium) with impact 0.7170 or (MFCC 3 is medium) with impact 0.7650 or (MFCC 4 is medium) with impact 1.0000 or (MFCC 5 is high) with impact 0.7429 or (MFCC 6 is high) with impact 0.7157 or (MFCC 7 is high) with impact 0.7257 or (MFCC 8 is high) with impact 0.7170 or (MFCC 9 is high) with impact 0.7164 or (MFCC 10 is very high) with impact 0.7196 or (MFCC 11 is very high) with impact 0.7155 or (MFCC 12 is high) with impact 0.7218 or (MFCC 13 is small) with impact 0.7326 then (sound identification is normal heart sound) with certainty 0.7191.

Rule 3—If (mean value is small) with impact 0.7192 and (median value is small) with impact 0.7152 and (standard deviation is high) with impact 0.7527 and (mean absolute deviation is high) with impact 0.7257 and (quantile 25 is very small) with impact 0.7153 and (quantile 75 is high) with impact 0.7153 and (signal IQR is high) with impact 0.7153 and (sample skewness is medium) with impact 0.7248 and (sample Kurtosis is medium) with impact 0.7167 and (signal entropy is high) with impact 0.7210 and (spectral entropy is medium) with impact 0.7410 and (dominant frequency value is medium) with impact 0.7283 and (dominant frequency magnitude is medium) with impact 0.7430 and (dominant frequency ratio is medium) with impact 0.7210 and (MFCC 1 is small) with impact 0.7255 and (MFCC 2 is medium) with impact 0.7170 and (MFCC 3 is small) with impact 0.7650 and (MFCC 4 is very small) with impact 1.0000 and (MFCC 5 is small) with impact 0.7429 and (MFCC 6 is small) with impact 0.7157 and (MFCC 7 is small) with impact 0.7257 and (MFCC 8 is small) with impact 0.7170 and (MFCC 9 is small) with impact 0.7164 and (MFCC 10 is small) with impact 0.7196 and (MFCC 11 is small) with impact 0.7155 and (MFCC 12 is medium) with impact 0.7218 with impact 0.7326 and (MFCC 13 is medium) then (sound identification is normal heart sound) with certainty 0.1631.

Rule 4—If (mean value is very small) with impact 0.7192 or (median value is very small) with impact 0.7152 or (standard deviation is small) with impact 0.7527 or (mean absolute deviation is medium) with impact 0.7257 or (quantile 25 is very small) with impact 0.7153 or (quantile 75 is small) with impact 0.7153 or (signal IQR is high) with impact 0.7153 or (sample skewness is small) with impact 0.7248 or (sample Kurtosis is medium) with impact 0.7167 or (signal entropy is medium) with impact 0.7210 or (spectral entropy is very small) with impact 0.7410 or (dominant frequency value is small) with impact 0.7283 or (dominant frequency magnitude is small) with impact 0.7430 or (dominant frequency ratio is very small) with impact 0.7210 or (MFCC 1 is small) with impact 0.7255 or (MFCC 2 is small) with impact 0.7170 or (MFCC 3 is small) with impact 0.7650 or (MFCC 4 is very small) with impact 1.0000 or (MFCC 5 is very small) with impact 0.7429 or (MFCC 6 is very small) with impact 0.7157 or (MFCC 7 is very small) with impact 0.7257 or (MFCC 8 is very small) with impact 0.7170 or (MFCC 9 is small) with impact 0.7164 or (MFCC 10 is small) with impact 0.7196 or (MFCC 11 is small) with impact 0.7155 or (MFCC 12 is small) with impact 0.7218 or (MFCC 13 is medium) then with impact 0.7326 (sound identification is normal heart sound) with certainty 2.7535.

## 5. Conclusions

The evolving fuzzy neural network model proposed in this paper obtained better results than state-of-the-art for online training. The generated fuzzy rules represent the model’s transparency in solving complex problems, and at the same time, determining logical and linguistic relationships to facilitate feedback regarding how the model arrived at the final results. This approach proved to be promising because it managed to deal with a complex problem in an organized and coherent way, extracting knowledge about the data submitted to the model. A positive point to be observed was the powerful performance of models based on online training, surpassing by three percent the accuracy of the state-of-the-art model under the same established test conditions. As a weakness of this model, a long time was observed in defining the weights of neurons in the first layer, mainly in training in evolution. A non-parametric model’s advantage brings the complexity of this simple model to solve the problem, requiring no additional procedures to solve the problem. Only the choice of fuzzy neuron defines the element to be used in solving the problem. It should be noted that some models, especially those with greater flexibility in the use of t-norm and t-conorm operators, performed better concerning accuracy. The hybrid model proposed in this paper has good prospects for working with new cases of the same nature, allowing new cases to be analyzed and with a 90% possibility of identifying problems related to heart murmurs. The extracted fuzzy rules can help professionals quickly identify patterns and adopt specific treatments to attack existing patterns in the patient’s diagnosis.

Future work related to this paper can be related to working with different datasets on heart murmurs, and tests with other models. It is also advisable to create other evolving fuzzy neural network architectures with different fuzzification approaches, defuzzification and training. Another future approach to the model would be to work on the prediction of future treatments such as those addressed in the work of Jeong et al. [59], which addresses blood pressure problems.

Forthcoming efforts may also be carried out as an expansion of this work acting dynamically in evaluating other datasets available in the literature, appraising other features, types of heart problems caused by a heart murmur or other types of diseases that can be collected through the same procedures that originated the dataset covered in this paper. An advantage of the proposed model is that it can act dynamically with numerical data of any problem, such as in the medical area, generating fuzzy rules adapted as new information is collected. The (linguistically readable) fuzzy rules generated from this use of the model can support and facilitate routine treatments, early identification of problems and other comorbidities commonly treated in hospitals and addressed in cutting-edge research.

## Figures and Tables

**Figure 1 sensors-20-06477-f001:**
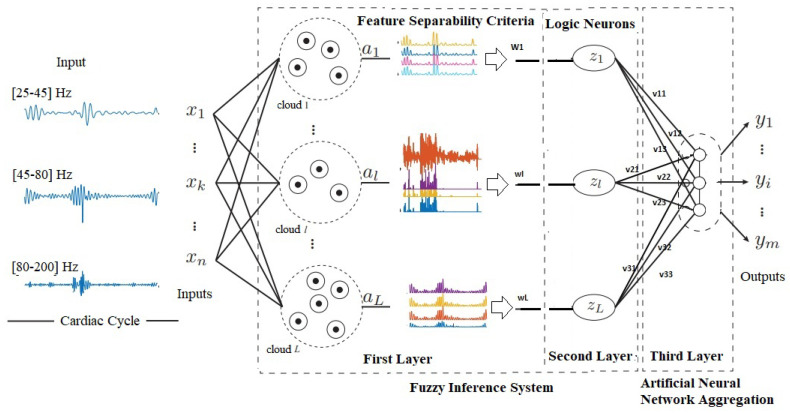
Evolving fuzzy neural network architecture.

**Figure 2 sensors-20-06477-f002:**
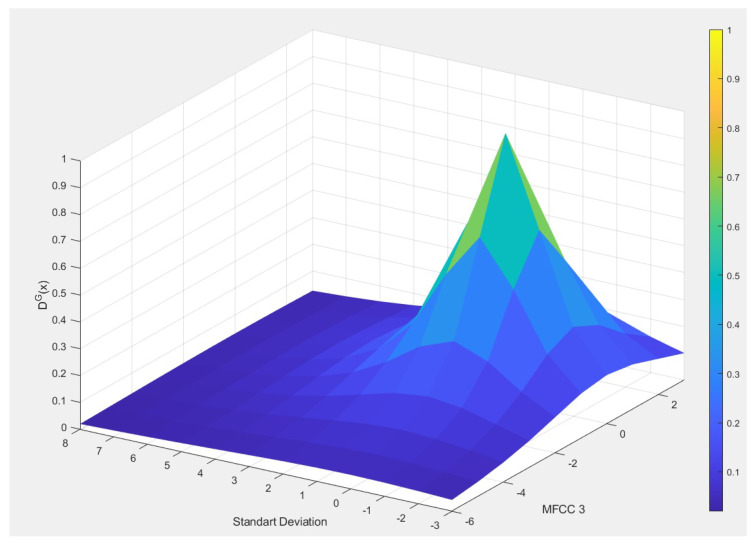
Global density visualization between standard deviation and MFCC 3; in this case one Gaussian neuron would be sufficient to describe the density distribution adequately.

**Figure 3 sensors-20-06477-f003:**
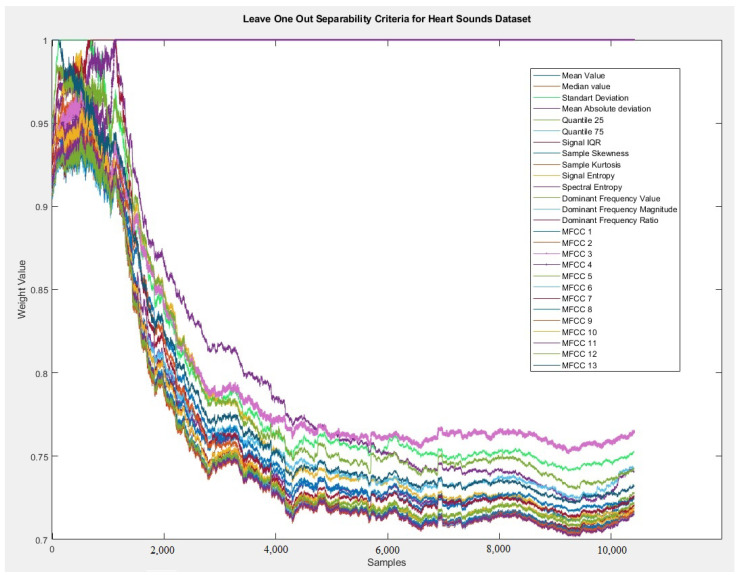
LOFO approach applied to the heart sounds dataset—please mind the change in the weight trends after around 1000 samples have been presented to the algorithm, where it crystallizes out that MFCC 3 and 4 are dominant features compared to all others (at the beginning all are nearly equally important).

**Figure 4 sensors-20-06477-f004:**
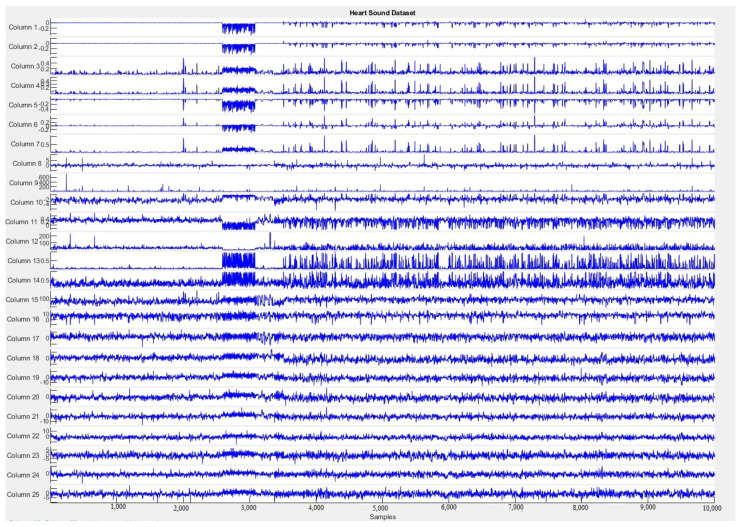
Heart sounds dataset features over time.

**Figure 5 sensors-20-06477-f005:**
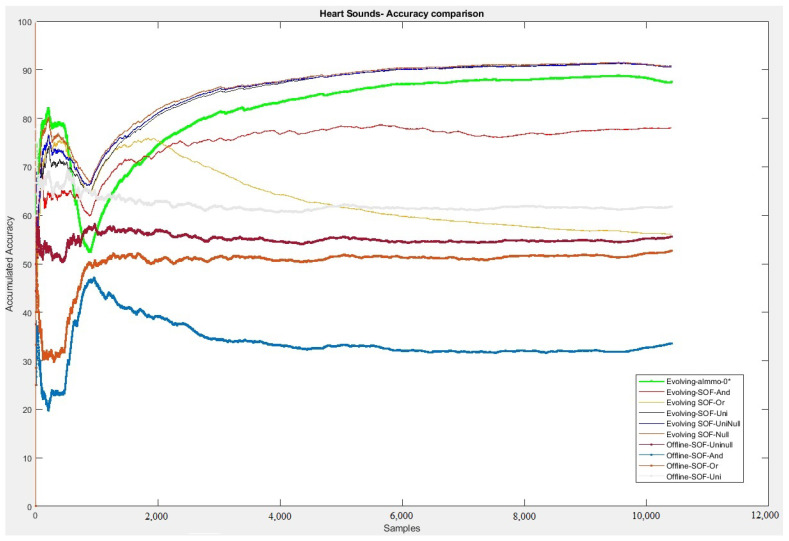
Heart sounds dataset—accumulated accuracy over the whole data stream.

**Table 1 sensors-20-06477-t001:** Relationship between g, u and β and the respective fuzzy operators.

*g*	*u*	β	Fuzzy Operator	Fuzzy Neuron
0	0	0	t-conorm (OR)	UNI-neuron
0	0	1	t-conorm (OR)	NULL-neuron
0	1	0	t-norm (AND)	AND-neuron
0	1	1	t-conorm (OR)	OR-neuron
1	0	0	t-conorm (OR)	OR-neuron
1	0	1	t-norm (AND)	AND-neuron
1	1	0	t-norm (AND)	NULL-neuron
1	1	1	t-norm (AND)	UNI-neuron

**Table 2 sensors-20-06477-t002:** Input features of the cardiac sounds dataset [4].

Dataset Dimensions	Mean	Standard Deviation	Max	Min
Mean Value	−0.0131	0.0396	0.1696	−0.4726
Median Value	−0.0134	0.0399	0.2364	−0.4744
Standard Deviation	0.0824	0.0659	0.6126	0.0021
Mean Absolute Deviation	0.0484	0.0510	0.5283	0.0009
Quantile 25	−0.0413	0.0657	0.1212	−0.5731
Quantile 75	0.0144	0.0519	0.5727	−0.3869
Signal IQR	0.0557	0.0875	1.0888	0
Sample Skewness	−0.1052	0.8711	10.3526	−6.2164
Sample Kurtosis	17.3596	15.8936	754.0798	1.5330
Signal Entropy	−1.8492	0.7922	0.6655	−6.8675
spectral entropy	0.2589	0.1994	0.7812	−0.3333
dominant frequency value	26.8757	26.0346	254.0303	0
Dominant Frequency Magnitude	0.1472	0.2091	1.0000	0.0102
Dominant Frequency Ratio	0.3399	0.2023	1.0000	0.0184
MFCC 1	96.8203	5.4674	118.0333	77.2747
MFCC 2	6.9214	4.3758	17.7092	−15.3445
MFCC 3	1.3577	3.6801	13.9213	−15.4105
MFCC 4	−2.0712	3.8150	14.8043	−14.6614
MFCC 5	−2.0186	3.4275	15.6168	−17.0746
MFCC 6	−2.1773	3.1318	13.6594	−14.6325
MFCC 7	−1.9079	2.7580	14.5581	−16.6508
MFCC 8	−1.7505	2.5032	17.4436	−14.6091
MFCC 9	−1.4809	2.2581	9.3733	−11.8959
MFCC 10	−1.3294	2.2276	10.3163	−14.1284
MFCC 11	−1.0183	1.9913	8.2289	−10.8879
MFCC 12	−1.0138	1.8840	12.0782	−16.5322
MFCC 13	−1.0221	1.6583	6.6063	−10.5979
Class	0.7574	0.4287	1.0000	0

**Table 3 sensors-20-06477-t003:** Variants of logical neurons and their construction.

Neuron	Neuron Representation	Fuzzy Logic Operator	Relevancy Transformation	Reference
AND	${\vec{Z}}_{rul}=AND\left(w;a\right)=T_{i=1}^{n}\left(w_{i}\;s\;a_{i}\right)$	T (product)	-	[14]
OR	${\vec{Z}}_{rul}=OR\left(w;a\right)=S_{i=1}^{n}\left(w_{i}\;t\;a_{i}\right)$	S (probabilistic sum)	-	[14]
UNI	${\vec{Z}}_{rul}=UNI\left(w;a\right)=U_{i=1}^{n}p\left(w_{i},a_{i}\right)$	$U\left(x,y,g\right)=\left\{\begin{array}{c} g\;T\left(\frac{x}{g},\frac{y}{g}\right),if\;\;y\;\in \left[0,g\right]\\ g+\left(1-g\right)\;S\;\left(\frac{x-g}{1-g},\frac{y-g}{1-g}\right),\;if\;y\;\in \;\left(g,1\right] \end{array}\right.$	$p\left(w;a;g\right)=wa+\bar{w}g,$	[40]
NULL	${\vec{Z}}_{rul}=NULL\left(w;a\right)=N_{i=1}^{n}p\left(w_{i},a_{i}\right)$	$NU\left(x,y,u\right)=\left\{\begin{array}{c} uS\left(\frac{x}{u},\frac{y}{u}\right),if\;\;y\;\in \left[0,u\right]\\ u\;+\;\left(1-u\right)\;T\;\left(\frac{x-u}{y-u},\frac{1-y}{1-u}\right),if\;\;y\;\in \;\left[u,1\right] \end{array}\right.$	$p\left(w,a,u\right)=wa+\bar{w}u$	[41]
UNINULL	${\vec{Z}}_{rul}=UNI^{NUL}\left(w,a,\beta ,g,u\right)=N^{un_{i=1}^{n}}p\left(w_{i},a_{i}\right)$	$N^{un}\left(x,y,\beta ,g,u\right)=\left\{\begin{array}{c} \beta U_{1}\left(\frac{x}{\beta },\frac{y}{\beta }\right),\;if\;x,y\;\in \left[0,\beta \right]\\ \beta \;+\;\left(1-\beta \right)\;U_{2}\;\left(\frac{x-\beta }{1-\beta },\frac{y-\beta }{1-\beta }\right),\;if\;x,y\;\in \;\left[\beta ,1\right] \end{array}\right.$	$p\left(w,a,\beta ,g,u\right)=\left\{\begin{array}{c} wa+\bar{w}\frac{g}{\beta },\;if\;U_{1}\\ wa+\bar{w}\frac{u-\beta }{1-\beta },\;if\;U_{2} \end{array}\right.$	[42]

**Table 4 sensors-20-06477-t004:** Related approaches applied to the cardiac sound dataset.

Author	Model	Accuracy/Macc	Reference
Potes et al.	CNN	0.8602	[45]
Zahibi et al.	SVM	0.8509	[46]
Dominguez-Morales et al.	DNN.	0.9416	[47]
Latif et al.	RNN.	0.9861	[48]
Whitaker et al.	SVM	0.8926	[49]
Xiao et al.	CNN	0.9300	[50]
Deng et al.	MFCC-CRNN	0.9701	[51]
Shukla et al.	HMM	0.9807	[52]
Soares et al.	ALMMo-0 *	0.9304	[10]
Zeng et al.	TQWT-VMD-NN	0.9789	[53]
Chen et al.	CNN	0.9391	[54]
Aziz et al.	SVM	0.9524	[55]
Chowdhury et al.	DNN	0.9710	[56]
Li et al.	TWSVM	0.9040	[57]
Noman et al.	MSAR	0.8610	[58]

**Table 5 sensors-20-06477-t005:** Interpretability with respect to the (degree of) changes in fuzzy neurons during the evolution phase with new incoming data samples.

Rule 1 did change in 27 membership functions and by a degree of 0.7896 with consequent change from Normal Heart Sound to Heart Murmur.
Rule 2 did change in 27 membership functions and by a degree of 1.0000 with consequent change from Heart Murmur to Normal Heart Sound.
Rule 3 did change in 27 membership functions and by a degree of 1.0000 with no consequent change.
Rule 4 did change in 27 membership functions and by a degree of 1.0000 with no consequent change.

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
