# Peer review of "Identification of Heart Sounds with an Interpretable Evolving Fuzzy Neural Network"

_sensors, 2020, doi:10.3390/s20226477_

Round 1

Reviewer 1 Report

* (line 136) Authors said that the number of neurons is automatically determined. There is a highly related technique, 'neural architecture search (NAS),' that the authors must check out. What would be the major difference between NAS and this work?

* (line 144) Authors insist that random initialization of weight parameters does not give comprehensibility, and propose to use feature importances as weight values. However, random initialization is not the reason for incomprehensibility. When we design neural networks, we just determine the number of (hidden layer) neurons but do not know what the neurons mean, and this is the main reason of the incomprehensibility. If we define each neuron manually, then it will be interpretible even if its values are randomly initialized because their weights will be estimated based on training data. Authors must check this point carefully.

* (line 224) Authors use a new term here 'Leave One Out Separability Criterion,' but use another term 'LOFO' at line 237. It seems that line numbers are missing between 236 and 237. I saw the initial mention of Separability Criterion in line 63, but there is no clear definition of Separability Criterion, and relationship between Separability Criterion and LOFO.

* (line 428, Table 4) It seems that RNN and HMM models better than the others. This might be explained that RNN and HMM benefit from their special capability of modeling sequential patterns because the data of this paper is essentially sequences.

* (line 439) I do not agree that some models are not capable of on-line training with larger dataset. Basically, all neural network models can be trained in on-line fashion. But a big limitation of neural network models is that they are mainly end-to-end architecture, so their results are not interpretible. So I recommend you to take the interpretibility as the biggest contribution, and you might need to prove the effectiveness of 'evolving mechanism' by additional experiments with/without 'evolving mechanism.'

* (line 439) There is a recently published work that deals with 'blood-pressure streams' in online one-by-one manner. Surprisingly this work predicts 'future(e.g., 3 mins ahead) streams' in real-time. You may need to refer it and suggest your future work related to it.

- "Prediction of Blood Pressure after Induction of Anesthesia Using Deep Learning: A Feasibility Study," Applied Sciences, 2019.

Author Response

Dear Journal Sensors reviewers and technical team.

First of all, we would like to thank you for the valuable comments on the paper submitted to this esteemed journal, and we also thank you for the technical contributions in the efficient suggestions provided in this paper. Below, all questions were adequately explained and, when incorporated into the text, are highlighted in red.

We have made adequate changes to the original manuscript in order to address your suggestions and comments.

We hope that the new version of the manuscript meets your expectations.

Kind regards.

The authors.

Dear reviewer. Thanks for the comments about our work. Below are the considerations necessary to answer your valuable questions.

Comment #1:

* (line 136) Authors said that the number of neurons is automatically determined. There is a highly related technique, 'neural architecture search (NAS),' that the authors must check out. What would be the significant difference between NAS and this work?

Answer: Many thanks for indicating this related approach. The two approaches were appropriately compared in Section 2.1 (see a paragraph in red font there), explaining the differences between the techniques used therein. One defines its architecture to improve outputs based on different optimization techniques. Ours defines the automatic architecture to represent the input data using terms capable of transforming the problem into an interpretable view. Furthermore, our model is adaptive and evolving in that it can be updated with single new samples (e.g., from a stream or additional data set later gathered, etc.) without requiring re-training from scratch. 

Comment #2:

* (line 144) Authors insist that random initialization of weight parameters does not give comprehensibility and propose to use feature importances as weight values. However, random initialization is not the reason for incomprehensibility. When we design neural networks, we determine the number of (hidden layer) neurons but do not know what the neurons mean, and this is the main reason of the incomprehensibility. If we define each neuron manually, then it will be interpretible even if its values are randomly initialized because their weights will be estimated based on training data. Authors must check this point carefully.

Answer: many thanks for pointing out this interesting aspect. What we meant is that once you have a random step in your approach, for different training runs (with the same data set), typically you will end up with different' results' (learned weights and thus also learned neurons in this case) => non-deterministic results; this is mostly also the case when you use the randomness just in the initialization because optimization-based (gradient-based) learning techniques do not necessarily find the global (unique) optima of the non-linear (neuron) weights (a still unresolved research issue), as long as the learning problem is not too trivial (as is the case for multi-dim real-world problems). That means that their found (typically local) solutions depend on the initialization. However, the non-deterministic results in using any random steps make a model challenging to interpret. Our approach for eliciting feature weights, which are included as neuron weights in their construction, is a deterministic one (always leading to the same weights when sending the same data set into the learning process). That achieves interpretability over different training runs. We made this better clear in the text.

Comment #3:

* (line 224) Authors use a new term here 'Leave One Out Separability Criterion,' but use another term 'LOFO' at line 237. It seems that line numbers are missing between 236 and 237. I saw the initial mention of Separability Criterion in line 63, but there is no clear definition of Separability Criterion, and relationship between Separability Criterion and LOFO.

Answer: many apologies for this confusion; when we say (leave one out) separability criterion, we always mean leave-one-feature-out separability criterion. Thus, it should be LOFO = Leave-One-Feature-Out separability criterion to avoid confusion with the leave-one(-sample-)out approach in the cross-validation sense. We checked the whole manuscript carefully and corrected the terminus wherever stated wrongly.

Comment #4:

* (line 428, Table 4) It seems that RNN and HMM models better than the others. This might be explained that RNN and HMM benefit from their special capability of modeling sequential patterns because the data of this paper is essentially sequences.

Answer: Many thanks for the indication. This information raised was duly included in the text after the table, as mentioned earlier. Both approaches indeed act on time-series data (by respecting the time lags in the layers reconstruction), but they establish an off-line model (based on pre-collected time-series), which cannot be updated further with data. We are proposing an evolving adaptive approach, which is initially trained with a couple of samples only and then further updated on demand and the fly with new incoming samples. That results in the accuracy trend lines over time, as shown in Figure 5 (there is also a fair comparison with other adaptive, evolving approaches). The end accuracy achieved by our approach after the whole stream is used for comparison purposes with RNN and HMM, which were trained on the whole data stream at once (so, they see all the data in advance). In this sense, their higher accuracies are somewhat expected. However, despite this fact, still, they are not so much outperforming our approach (and other off-line methods even less), which we want to show in Table 4 (as we have only accuracy numbers for these methods when using the whole data stream at once in the respective papers). We discussed this further in Section 4.3.

Comment #5:

* (line 439) I do not agree that some models are not capable of on-line training with larger dataset. Basically, all neural network models can be trained in on-line fashion. But a big limitation of neural network models is that they are mainly end-to-end architecture, so their results are not interpretible. So I recommend you to take the interpretibility as the biggest contribution, and you might need to prove the effectiveness of 'evolving mechanism' by additional experiments with/without 'evolving mechanism.'

Answer. Many thanks for this argument. The statement is based on updating parameters off-line, that is, within a context of dynamic update of patterns. We agree that several models of neural networks can act on-line. However, how these models can update parameters may be inefficient or not coherent for immediate responses, fundamental to problems so necessary for immediate responses about treatment. The main advantage of on-line training is the single-pass, often recursive (speedy) updating of parameters. Many models with backpropagation-based training and related optimization techniques become inefficient when not considering some data previously seen (e.g., in forms of sliding windows); however, this requires past data revisited and often additional parameter steering the number of past samples. The results show that the rules (in this case, the neurons) initially defined are limited. In the online assessment of the problem, the results prove that the online models work better in the correct definition of treatment as new cases and data are evaluated. To prove the inefficiency of off-line trained models without evolving mechanisms included, their results were included in the evaluation, mostly see the new accuracy trend lines in Figure 5, which are termed in the legend as 'Offline-SOF.' The emphasis of our approach is in fact' interpretability,' as also pointed out in the title and the abstract and also early in the introduction. Finally, we think that the combination of model interpretability and the on-line, adaptive evolving nature of our approach is a new appealing aspect.

Comment #6:

* (line 439) There is a recently published work that deals with 'blood-pressure streams' in on-line one-by-one manner. Surprisingly this work predicts 'future(e.g., 3 mins ahead) streams' in real-time. You may need to refer it and suggest your future work related to it.

- "Prediction of Blood Pressure after Induction of Anesthesia Using Deep Learning: A Feasibility Study," Applied Sciences, 2019.

Answer. Thanks for the excellent suggestion. This proposal for a future approach is exciting. The work was appreciated and duly included in the conclusions section, which refers to future paper approaches.

Reviewer 2 Report

Dear Authors,

I have some comments on your article:

  1. At the beginning of section 3. Materials and Methods, it would be good to specify which diagnostic equipment the registered data is being processed.
  2. Descriptions in Figure 1. too small. Especially the description of the input data.
  3. Table 1. It seems to me that the reversal of this table is not good. Please, maybe format it differently and let it be placed rather horizontally.
  4. All indexes in symbols in text and equations should be checked carefully.
  5. Alternatively, you can write something more about the summary of the method's versification on databases from other data and how to implement this algorithm for medical diagnostics.

Author Response

Dear Journal Sensors reviewers and technical team.

First of all, we would like to thank you for the valuable comments on the paper submitted to this esteemed journal, and we also thank you for the technical contributions in the efficient suggestions provided in this paper. Below, all questions were adequately explained and, when incorporated into the text, are highlighted in red.

We have made adequate changes to the original manuscript in order to address your suggestions and comments.

We hope that the new version of the manuscript meets your expectations.

Kind regards.

The authors.

Dear reviewer. Thanks for the comments about our work. Below are the considerations necessary to answer your valuable questions.

Dear Authors,

I have some comments on your article:

Dear reviewer. Thanks for the comments. We believe that your considerations have made the paper more coherent and understandable to the reader's view.

Comment #1:

At the beginning of section 3. Materials and Methods, it would be good to specify which diagnostic equipment the registered data is being processed. 

Response: 

 Answer: Thank you very much for your suggestion.

The information was included in the text in the appropriate section.

Comment#2: 

The descriptions in Figure 1 are too small—especially the description of the input data.

Answer: Thanks for the consideration. The fonts in the figure have all been changed to a larger font to increase readability. 

Comment #3:

Table 1. It seems to me that the reversal of this table is not good. Please, maybe format it differently and let it be placed rather horizontally. 

Answer: Thanks for the consideration. The table was changed to the suggested form.

Comment #4: 

All indexes in symbols in text and equations should be checked carefully.

Answer: Many thanks for indicating this omission. There were some wrong indices and (partial) terms in the formulas, which we all corrected upon a careful checking of all formulas.

Comment #5: 

Alternatively, you can write something more about summarizing the method's versification on databases from other data and how to implement this algorithm for medical diagnostics. 

Answer: Many thanks for this valuable contribution. Indeed, the model proposed in this paper can act in several areas where the data collected are numerical. Therefore, the necessary explanations were included in future works reported in the Conclusion section of the paper.

Round 2

Reviewer 1 Report

Authors answered all the comments appropriately, and revised the manuscript.

Reviewer 2 Report

Dear Authors,

Thank you very much for introducing changes that have improved the quality of the article. I have no more comments.

Best regards